# Evaluation of cattle farmers' knowledge, attitudes, and practices regarding antimicrobial use and antimicrobial resistance in Rwanda

Elise M. Hirwa[1☯], Gisele Mujawamariya[1☯], Natnael Shimelash[2], Anselme Shyaka[1]*

1 Center for One Health, University of Global Health Equity, Kigali, Rwanda, 2 Simulation and Skills Center, University of Global Health Equity, Kigali, Rwanda

☯ These authors contributed equally to this work.
* ashyaka@ughe.org, shyakaa@gmail.com

**Data Availability Statement:** All relevant data are within the manuscript and its Supporting information files.

## Abstract

The misuse of antimicrobials in livestock may lead to the emergence and spread of resistant pathogens harmful to human, animal, and environmental health. Therefore, determining the behavior and practices of farmers regarding antimicrobial use (AMU) and antimicrobial resistance (AMR) is crucial for addressing the growing threat of AMR. This cross-sectional study was conducted in the Eastern Province of Rwanda on 441 participants using a structured questionnaire to determine the knowledge, attitudes, and practices (KAP) towards AMU and AMR. Frequency distributions, chi-square test of association and logistic regression model were used to analyze the data. This study showed poor biosecurity measures at the farm level with various antimicrobials used here; 83.9% of participants obtained them from friends and neighbors and 61.9% used them for growth promotion. Our assessment revealed a low level of KAP towards AMR among cattle farmers from the study districts. Our data showed that at a 69% cutoff, only 52.6% of farmers had correct knowledge, whereas 56% had good attitudes (47% cutoff). Finally, 52.8% had correct practices toward AMR based on a calculated cutoff of 50%. Positive attitudes, correct knowledge, and practices regarding AMU and AMR were associated with higher educational levels. Sex was correlated with knowledge and attitudes, whereas farm location was associated with attitudes and practices. Farmers expressed a need for more access to veterinary services and AMR-related training for themselves, the community animal health workers, and veterinarians. This study highlighted the low levels of KAP associated with using antimicrobials, which may lead to the misuse of antimicrobials and the spread of AMR. It is imperative to develop and implement cross-cutting measures to minimize antibiotic usage and reduce the risk of antibiotic resistance.

**Funding:** The author(s) received no specific funding for this work.

## Introduction

Antimicrobial resistance (AMR) is a significant global public health problem [1]. AMR refers to a situation in which microorganisms acquire the capacity to survive antimicrobials intended to kill them or stop their maturation and multiplication [2]. Antimicrobial misuse is an important factor that enables AMR in humans and animals because of prolonged exposure to antimicrobials and bacteria developing resistance [3]. AMR constitutes a global challenge for animals, humans, and the shared environment and thus requires a One Health approach [4]. For instance, resistant pathogens can be transmitted from humans to animals and vice versa through various routes, including the environment, direct contact, and/or food products [5, 6].

Although AMR is distributed worldwide, its significance is disproportionate among Low- and Middle-income countries (LMICs) [7]. In 2019, AMR was responsible for 4.95 million deaths in humans globally; the highest proportion was attributed to sub-Saharan Africa and other LMICs [1]. The AMR-associated burden and death toll are predicted to increase to 10 million annual global deaths by 2050 [8, 9], requiring global coordinated action [10, 11]. It is also forecasted that by 2050, AMR will contribute to an estimated economic loss of USD 100 trillion globally by reducing over seven percent of livestock production [9, 12], thus severely affecting food security. Despite this alarming situation, antimicrobial use (AMU) is projected to increase exponentially at the farm level [13]. A study conducted in 228 countries revealed that 63,151 tons of antimicrobials were used for livestock production in 2010 [14]. In 2013, antimicrobial consumption increased to approximately 131,000 tons, and this figure is estimated to peak at 200,235 tons by 2030 [14]. These results demonstrate the importance of responsible farm-level AMU under the supervision of qualified veterinarians. However, poor access to veterinary services, especially in remote areas, remains a predominant contributor to antimicrobial misuse and unregulated distribution of antimicrobials [15], resulting in farmers self-prescribing and administering antimicrobials to their livestock [16].

A growing amount of evidence shows that high AMU by farmers is associated with their knowledge, attitudes, education level, and farming experience [17]. Studies conducted in East Africa on the knowledge, attitudes, and practices (KAP) of cattle farmers have highlighted the association between KAP and AMU at the farm level and the possible emergence of AMR [17–20]. Similar to other African countries, the AMR burden in Rwanda is of great importance. Resistant pathogens have been detected in humans [21], and food animals and their products [22–24]. Previous evaluations have highlighted the underperformance of veterinary services [25], which suggests a lack of access to veterinary care and services, leading to farmers performing treatment under inadequate supervision.

There is a lack of data on the AMR KAP of cattle farmers in Rwanda to inform decision-making concerning AMU and the prevention of the spread of AMR in livestock farms. Thus, the current study aimed to evaluate farmers' KAP regarding AMU and AMR in three districts located in East Rwanda.

## Methods

### Study setting

This study was conducted in the Eastern Province of Rwanda (Fig 1). It is the largest province in the country and is inhabited by over 3.5 million people [26]. Most areas are rural, with an estimated 77.7% of people relying on agriculture, especially animal husbandry, as their economic activity [26]. The province has the largest area of farmland, accounting for more cattle herds and higher milk production, making it a prospective agricultural hub for the country.

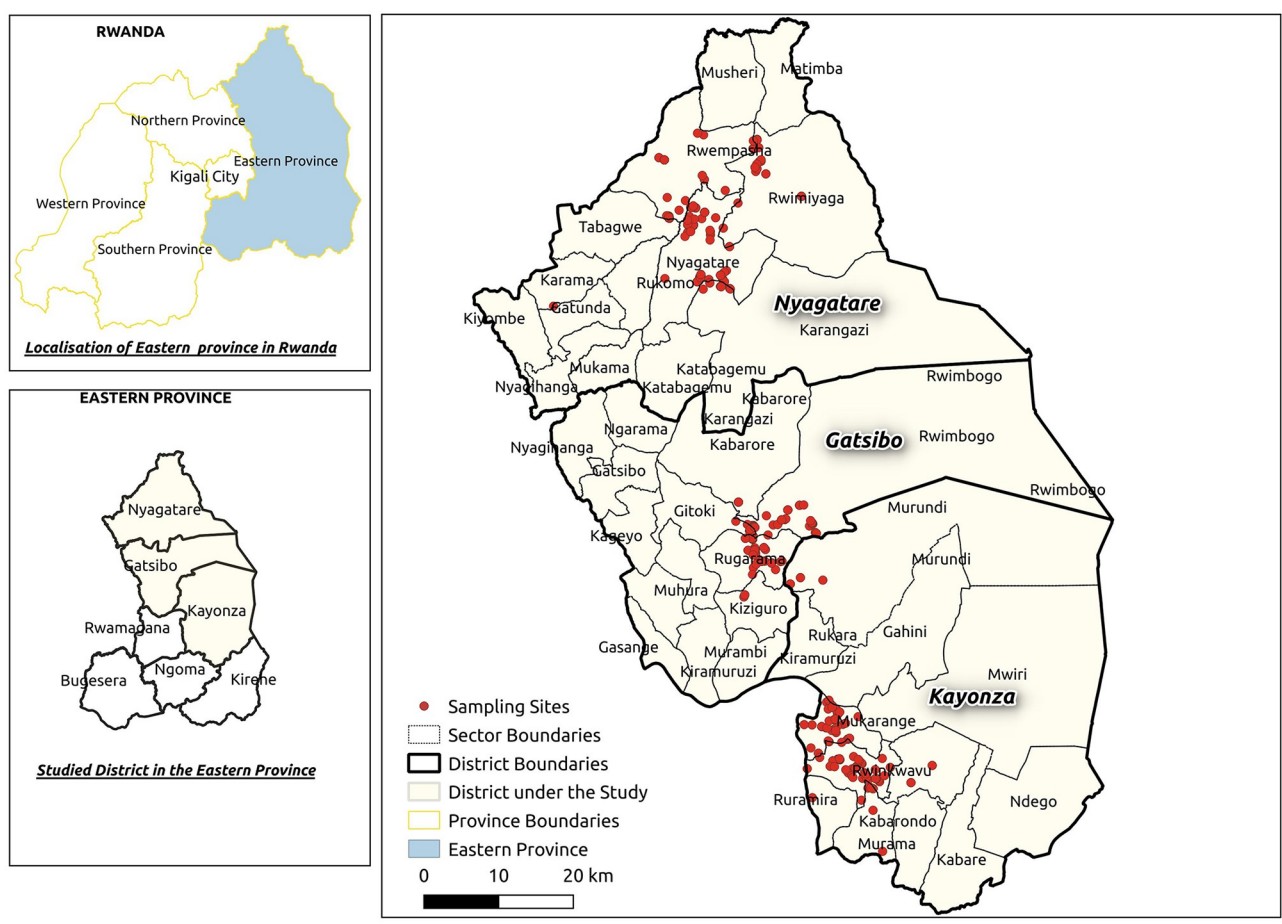

**Fig 1. Map of the study area with the interviewed farm locations (red points) in the Kayonza, Gatsibo, and Nyagatare districts.** This map was created using QGIS Ver. 3.20. The layers are freely accessible from https://www.diva-gis.org/datadown and they can be shared under CC-BY license 4.0.

The study participants were recruited in three districts, namely, Gatsibo, Kayonza, and Nyagatare, which have the largest cattle populations in the province and are Rwanda's primary milk sources [26].

## Study design

A cross-sectional survey was used to understand farmers' KAP regarding AMR and the drivers of AMU.

## Study population and sampling strategy

The target population for this study comprised cattle farmers aged 18 years and older living in the three districts. The sample size was calculated using an unlimited population and using the Cochrane formula indicated for categorical variables [27] in line with other similar AMR KAP. Hence, a proportion (47%) of farmers' knowledge of AMR was used from a similar AMR KAP study in Kenya [20], as there were no other precursor studies on KAP on AMR in Rwanda. To account for missing data and invalid entries, 15% of the 383 participants were added to the required sample size, rounding to 441 participants, selected using a multistage sampling method. First, informed by the District Animal Resources Officers (DAROs), provinces,

districts, and sectors were selected based on available livestock numbers. Second, from each district, two sectors were selected based on their higher number of cattle, for a total of six sectors. Finally, Excel's RAND function randomly selected three cells per sector (18 cells) and three villages per cell (54 villages). Eight farms from each village were randomly selected and recruited for interviews.

## Data collection tool and validation

This study was conducted between May and June 2022. Prior to the study, the district and sector offices were made aware of the research, and they granted approval. Four data collectors with academic backgrounds and experience in veterinary medicine, animal production, and data science collected the data.

Data was collected using the Kobo toolbox.

The research team designed and reviewed the data collection tool internally for content, understanding, and relevance. The tool was translated into Kinyarwanda by the team and reviewed by an animal health expert fluent in the language. The data collection tool was pretested three times on small and large farms located in Kigali City and Rwamagana District to assess the understanding of the content and to do reviews.

The tool contains Boolean questions, a Likert scale, and open-ended questions. The questionnaire comprised six parts (S1 Table). The first part entailed 14 questions addressing participants' demographics such as age, sex, education level, marital status, socioeconomic status, residential address, the role of the participant on the farm, type of farm, number of livestock on the farm, most prevalent livestock diseases as well as the frequency of visits by a veterinarian or a veterinary paraprofessional. The second part consisted of 13 yes/no, open-ended, and multiple-choice questions on water, hygiene, sanitation behaviors, practices, and biosecurity on farms. The third part contained 8 questions on AMU activities carried out on farms. The fourth part included 13 questions on the knowledge of antibiotic resistance and use. The answers to these questions were yes or no. It included questions on the level of understanding of AMR, antimicrobial residues, dosage, and dosage regimes. The fifth part comprised 9 questions on attitudes towards AMR. The questions were rated on a five-point Likert scale (strongly agree, agree, neutral, disagree, and strongly disagree). The sixth section included 12 questions on practices related to AMR and lastly, an open-ended question regarding possible measures to minimize the risk of AMR. After a literature review and validation by an animal health expert, the team developed a scoring rubric for the correct answers.

Data was submitted to Kobo servers daily and later downloaded and imported into SPSS for cleaning and analysis.

## Data analysis and management

"Data analyses were performed using SPSS Statistics version 27 (IBM Corp., Armonk, NY, USA) graphical user interface.

The demographics, knowledge levels, practice levels, attitude levels, and biosecurity-related items were summarized as counts and percentages using descriptive statistics.

The demographic characteristics included age category, gender, education level, socioeconomic status, marital status, district, participant role at the visited farms and the number of cattle owned.

Biosecurity-related items were viewed independently to create variables whose frequencies were calculated in the data analysis.

The three dependent variables were knowledge, attitudes, and practice (KAP) levels. Knowledge and practices were graded one point if correct and zero if incorrect, whereas, for

attitudes, the correct answer was graded as 2 points, neutral as 1 point, and incorrect answers were 0, based on the validated rubric. The total grades per participant were used to calculate the mean KAP score, which was used as a cutoff for "good" and "poor" knowledge, attitude, or practice. The calculated means were 69%, 47%, and 50% for knowledge, attitude, and practice, respectively. Participants with scores above or equal to the cutoff were considered to have good knowledge, attitude, or practice. In contrast, those with scores below the cutoff were considered to have poor knowledge, attitude, or practice.

Associations between demographics and KAP levels were analyzed using a chi-square test of association between KAP variables (Knowledge, attitudes, and practice levels) and potential explanatory independent variables (the demographic characteristics). The demographic factors that showed a significant association during the chi-square univariate analysis were used to perform a multivariable binary logistic regression analysis to identify the key independent variables affecting KAP toward AMR in the study area. The 5% level of significance was used to interpret the association results and the 95% confidence intervals for the adjusted odds ratios were used to evaluate the significance and direction of the associations.

## Results

### Demographics

Most participants were male (61.5%; Table 1) and the majority (40.1%) were aged between 46 and 65 years and the mean age was 47.7 years (SD = 16.4). More than half (54%) of the participants had a primary school education, and the vast majority (86.4%) were farm owners who owned between 1–3 cattle (62.1%) (Table 1, S2 Table).

### Biosecurity

Microbial diseases caused by viruses, bacteria, and fungi (34%) were the most common in the farms. Parasitic diseases also made up a high percentage (33%) of diseases stated as common in participants' farms (Fig 2, S3 Table).

Table 2 reviews the biosecurity levels of the participating cattle farms. Less than half of the population (48.5%) had water on their farms. More than a third travelled less than 15 minutes to reach water sources (39.9%). Most participants (38.3%) collected water from charged public wells, whereas 24.5% collected water from free water sources. Most participants (70.3%) reported that they cleaned their hands before performing cattle-related activities, by using, mostly, water, and soap (70.7%). Of the 441 study participants, 67.1% cleaned their cattle once a week, and 63.9% reported cleaning their barns at least daily, mostly (69.4%) by sweeping. Almost all farms (93%) did not have a foot bath at their entrances. However, 79.6% of the participants had fenced farms, and 67.3% of them reared animals within their farms without allowing the cattle to wander beyond their parameters (Table 2).

When participants were asked about the vaccines administered to their cattle, the majority had vaccinated their cattle against Lumpy skin disease (33%), followed by Rift Valley Fever (24%) and the Foot-and Mouth Disease (15%). Some did not have their cattle vaccinated (12%), or they had vaccinated them but did not know which vaccine it was (9%) (Fig 3).

### Knowledge of antimicrobial resistance and antimicrobial use

Participants' results regarding their knowledge of antimicrobial use and antimicrobial resistance are included in Table 3. More than half of the participants (52.8%; Table 3, S4 Table) did not know about AMR. The reported information source for those who knew about AMR was their own experiences on their farms (34.1%), whereas (31.3%) had heard it from

**Table 1. Socio-demographic data of cattle farmers.**

| Demographic data | Variables | Frequency (%) |
|---|---|---|
| Gender | Female | 170 (38.5) |
| | Male | 271 (61.5) |
| Age | 18–30 | 82 (18.6) |
| | 31–45 | 114 (25.9) |
| | 46–65 | 177 (40.1) |
| | >65 | 68 (15.4) |
| Education level | No education | 119 (27.0) |
| | Primary | 238 (54.0) |
| | Secondary | 76 (17.2) |
| | Tertiary | 8 (1.8) |
| Marital status | Single | 68 (15.2) |
| | Married | 317 (71.9) |
| | Widow/Widower | 49 (9.3) |
| | Divorced | 7 (1.6) |
| Socio-economic status | Category 1 | 46 (10.4) |
| | Category 2 | 183 (41.5) |
| | Category 3 | 212 (47.8) |
| Role of farmer | Owner | 381 (86.4) |
| | Worker | 60 (13.6) |
| Number of cattle owned | 1–3 | 274 (62.1) |
| | 4–6 | 54 (12.2) |
| | 7–9 | 24 (5.4) |
| | 10–12 | 18 (4.1) |
| | >12 | 71 (16.1) |
| District | Nyagatare | 152 (34.5) |
| | Gatsibo | 144 (32.7) |
| | Kayonza | 145 (32.9) |

veterinarians. When asked about the effect of antibiotic resistance, most participants (41.0%) replied that it killed cattle, whereas (21.8%) reported that it made it difficult to treat cattle. More than one-third of the participants chose animal products as the transmission mode of resistant pathogens from animals to humans. More than one-third of the patients were unaware of the transmission modes of resistant pathogens.

Almost all participants (90.9%) had used antimicrobials on their animals before. Most participants (40.6%) had used antimicrobials two to five times in the past 12 months, and many participants had antimicrobials prescribed (63.5%) and administered (50.1%) by veterinarians (Table 3).

Most participants (64%) had used antibiotics, with the majority using Tetracyclines (57.7%) and Betalactams (40.1%) (Figs 4 and 5).

While a third of participants (76.4%) used antimicrobials for treatment reasons, some (17%) stated using them for disease prevention in their cattle (Fig 6).

## Antimicrobial resistance

Table 4 summarizes the participants' knowledge-specific scores about AMR. More than half of the participants (51.7%) had good knowledge levels (Table 4, S5 Table). Most participants (68%) responded that antimicrobials could be used for the growth promotion of their cattle.

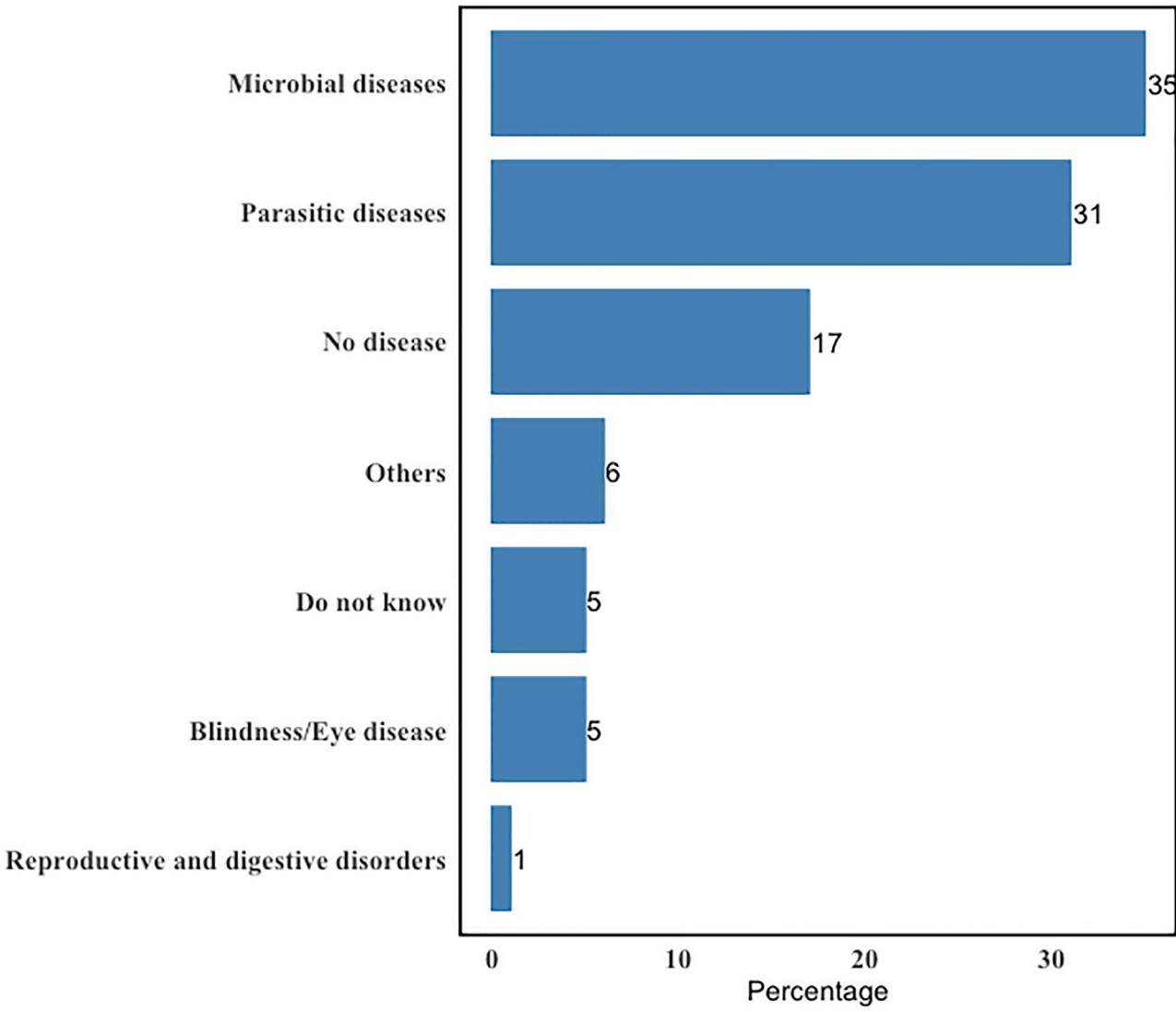

**Fig 2. Diseases mostly reported by the cattle farmers.**

More than a third of participants (38.8%) thought antimicrobials were food supplements. More than half of our participants knew that bacteria could develop resistance against antibiotics, however, the majority (80%) did not know that antimicrobials can show side effects after use.

The participant's attitude scores are summarized in Table 5 below. Their average attitude levels were low (less than 69%; Table 5, S6 Table). Nevertheless, most participants (57.8%) thought that AMR was a public health concern. Some (10.7%) did not agree that the withdrawal period should be respected before milking cattle. The majority (79.6%) affirmed that increased sales and distribution of antimicrobials would not reduce antibiotic resistance. Moreover, more than half of the participants (56.0%) agreed that they would treat their cattle themselves if the diseases did not get healed with veterinarians' treatment.

Practice-related scores are included in the Table 6. The average practice level was 60%. Close to half of the participants (48.1%) did not respect the course of antimicrobials, and more

**Table 2. Biosecurity level of the selected cattle farms in the study area.**

| Biosecurity related items | Variable | Frequency (%) |
|---|---|---|
| Is water available at your farm | Yes | 214 (48.5) |
| | No | 191 (43.3) |
| | Occasionally | 36 (8.2) |
| How long do you travel to reach a water source from your farm or stable | < 15 minutes | 176 (39.9) |
| | 15–30 minutes | 134 (30.4) |
| | > 30 minutes | 131 (29.7) |
| Where do you get the water you use in the farm from? | Spring | 68 (15.4) |
| | Free public well | 108 (24.5) |
| | Charged public well | 169 (38.3) |
| | Faucet | 60 (13.6) |
| | Surface water | 36 (8.2) |
| How many times per day do you clean your stable | Never | 20 (4.5) |
| | Once | 187 (42.4) |
| | More than once | 95 (21.5) |
| | Occasionally | 139 (31.5) |
| What do you normally use to clean your stable | Disinfectant | 9 (2.0) |
| | Sweeping | 396 (89.8) |
| | Water only | 9 (2.0) |
| | Water and soap | 7 (1.6) |
| | NAs | 20 (4.5) |
| Do you wash your hands before going in contact with cattle | Yes | 410 (93.0) |
| | No | 31 (7.0) |
| What do you normally use to clean your handz | Water only | 97 (22.2) |
| | Water and soap | 312 (70.8) |
| | Nas | 31 (7.0) |
| How many times do you clean your cattle per week | Never | 44 (10.0) |
| | Once | 296 (67.1) |
| | Twice | 89 (20.2) |
| | More than two times | 12 (2.7) |
| How do you dispose of the cow dung? | In my farm (pit) | 282 (63.9) |
| | I sell it | 3 (0.7) |
| | Open environment | 153 (34.7) |
| | I use it as fertilizer | 3 (0.7) |
| Does your farm have a fence | Yes | 90 (20.4) |
| | No | 351 (79.6) |
| Do your cows graze outside of the farm | Yes | 297 (67.3) |
| | No | 144 (32.7) |
| Does your farm have a foot bath at the entrance? | Yes | 430 (97.5) |
| | No | 11 (2.5) |

*The question collected information from 401 participants as 20 had replied "never" to the previous questions.
**Sweeping includes 83 participants who selected "other" such as fertilizer remover (removing cow dung).
***One entry was missing (N = 410 participants who reported to wash hands before being in contact with cattle)

than half (59.9%) kept antimicrobials for future use. Furthermore, the majority (83.9%) said they got their antimicrobials from friends and neighbors. Most participants (61.9%) also used antimicrobials for growth promotion on their farms, while more than half (53.7%; Table 6, S7 Table) used them for disease prevention.

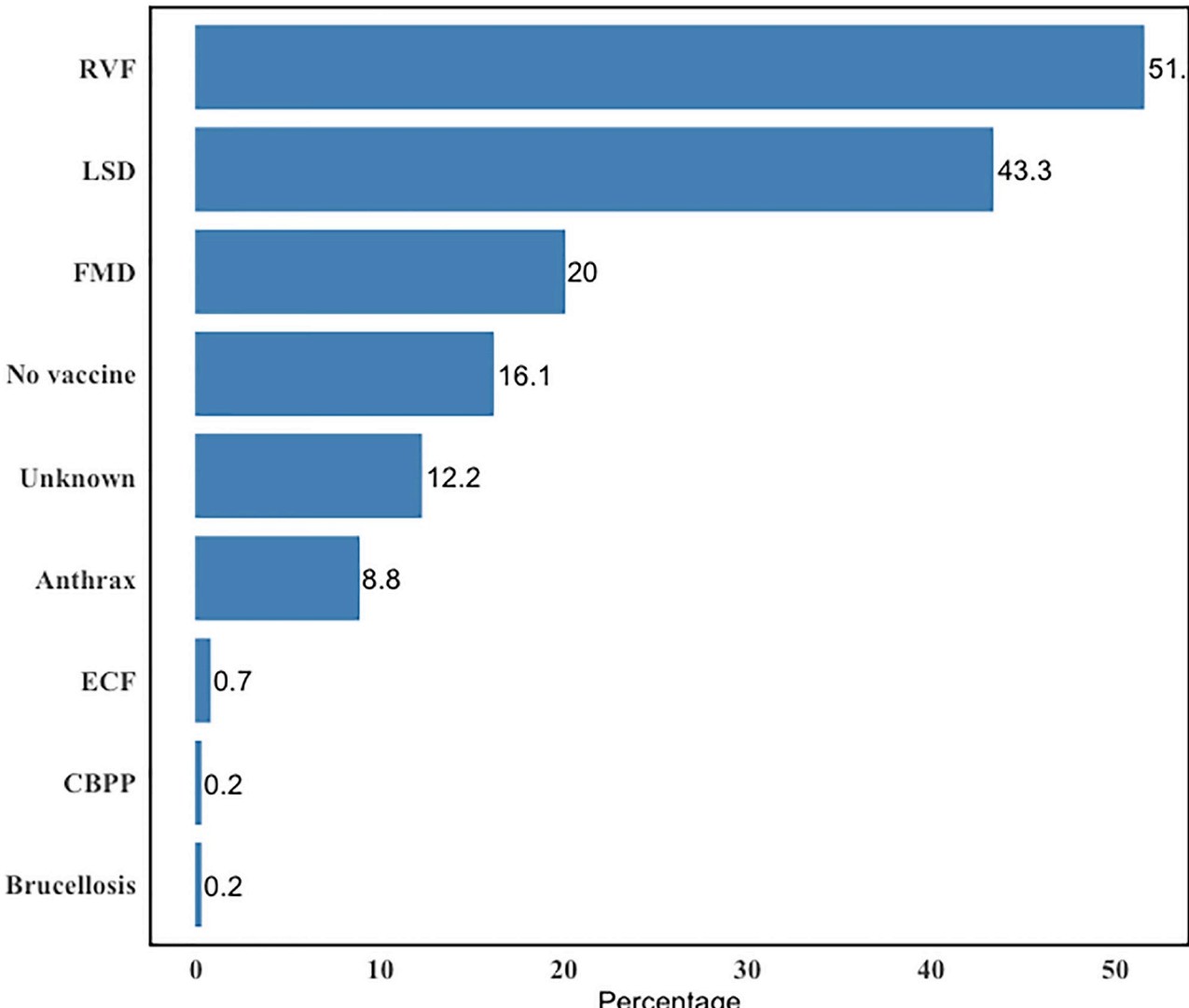

**Fig 3. Vaccines used by the cattle farmers.** FMD (Foot and Mouth Disease), RVF (Rift Valley Fever), LSD (Lumpy skin disease), ECF (East Coast Fever).

### Correlation between the KAP scores

Table 7 contains the results of the Pearson's correlation performed on the KAP scores. Knowledge and Attitude positively and moderately correlated (0.368, $p < 0.05$), whereas knowledge and practice were weakly correlated (0.080). The attitude score was significantly and positively correlated with the knowledge and practice scores (0.368 and 0.338) (Table 7).

### Factors associated with knowledge, attitudes, and practices outcomes

Knowledge levels were significantly associated with sex ($p < 0.001$), education level ($p = 0.003$), and the number of cattle owned ($p = 0.033$). Attitudes were significantly associated with education level ($p = 0.013$), district ($p = 0.001$), and the number of cattle owned ($p = 002$). Practice levels were significantly associated with sex ($p < .009$), marital status ($p = 0.047$), age ($p = 0.01$), the number of cattle owned ($p < 0.001$), and district ($p < 0.001$) (Table 8).

**Table 3. Antimicrobial use and general knowledge of AMR.**

| Antimicrobial use related questions | Variable | Frequency (%) |
|---|---|---|
| Have you ever used antibiotics? | Yes | 401 (90.9) |
| | No | 40 (9.1) |
| Who normally prescribes the antibiotics | A friend | 37 (8.4) |
| | Self | 84 (19.0) |
| | Veterinarians | 280 (63.5) |
| | NAs | 40 (9.1) |
| Who normally administers the antibiotics | A friend | 53 (12.0) |
| | Self | 127 (28.8) |
| | Veterinarians | 221 (50.1) |
| | NAs | 40 (9.1) |
| How many times have you used the antibiotics in the past 12 months? | Never | 14 (3.2) |
| | Once | 54 (12.2) |
| | 2–5 times | 179 (40.6) |
| | > 5 times | 150 (34.0) |
| Have you used antibiotics to promote growth in your cattle? | Yes | 219 (49.7) |
| | No | 182 (41.3) |
| | NAa | 40 (9.1) |
| Have you ever heard of antimicrobial resistance | Yes | 208 (47.2) |
| | No | 233 (52.8) |
| Where did you hear it from? | Friends | 13 (6.3) |
| | Media | 53 (25.5) |
| | I experienced it | 71 (34.1) |
| | Veterinarians | 65 (31.3) |
| | School | 5 (2.4) |

## Logistic regression analyses

The probability of being in the categories of good knowledge, attitudes, or practices, was determined using a multivariable binary logistic model. The odds of having good knowledge about AMR for those who had no formal education were 2.6 times (p = 0.03; Table 9) and those with primary level were 2.1 times (p = 0.07) higher than the odds of those who did secondary levels and above. In addition, farmers' sex was correlated with their knowledge of and attitudes towards AMR. In fact, the adjusted OR for having good knowledge and attitude were nearly two times higher in females compared to their counterparts for both knowledge and attitude scores (p = 0.07).

Participants' farm locations (districts) correlated with attitudes and practices but not knowledge. Specifically, looking at the adjusted OR, the farmers in Nyagatare and Gatsibo had 2.3 times and 1.7 times for adequate attitudes (p = 0.02 and p = 0.03) and 4.8 times and 4.9 times for good practices (p < 0.001), respectively, compared to farmers in Kayonza.

## Possible interventions

Most participants (73% male and 56% female) replied that training of farmers and veterinarians on AMR and AMU would be an important intervention for risk reduction of AMR at the farm level. This was followed by having veterinarians provide primary care (49% male and 35% female) (Fig 7).

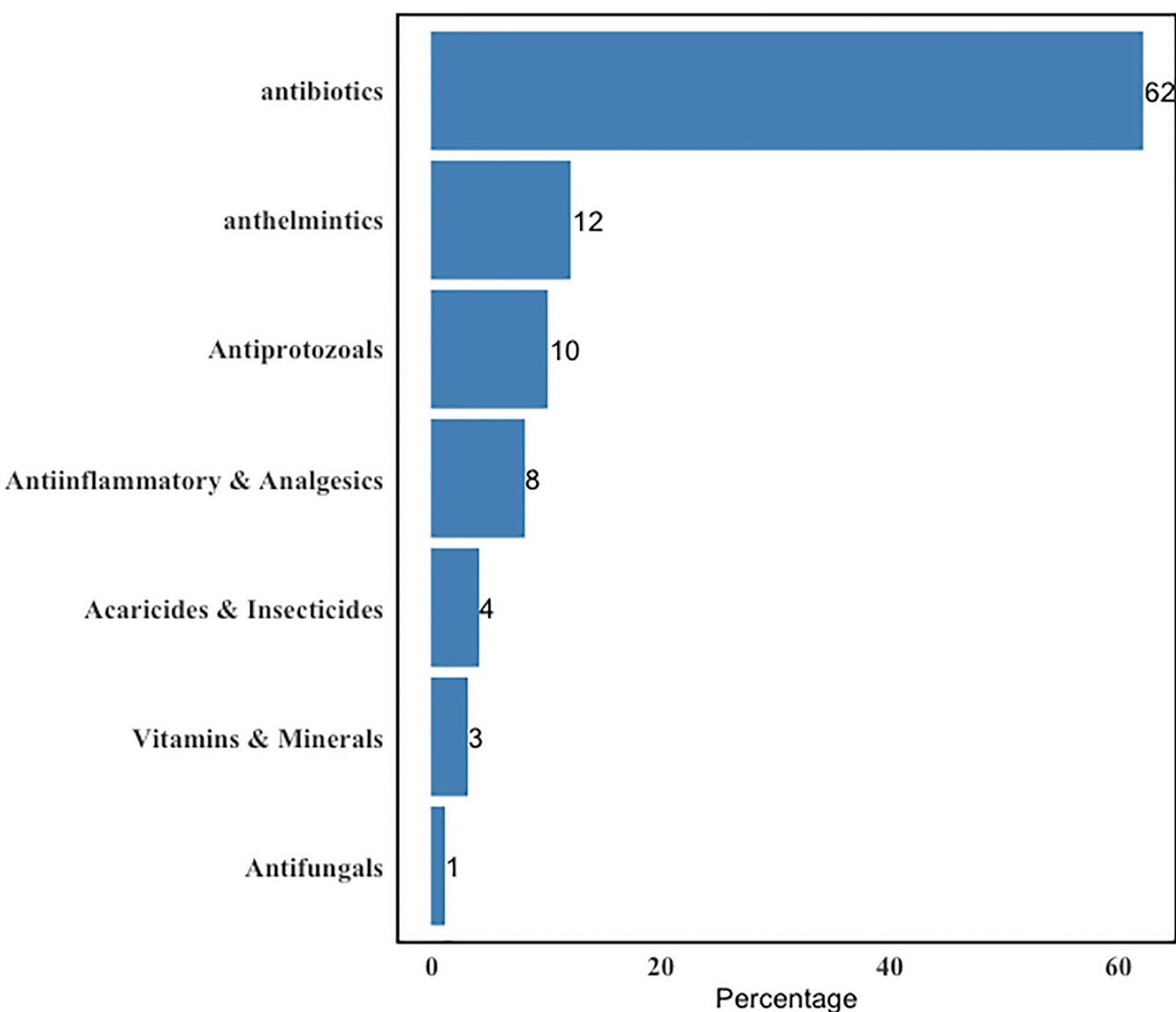

**Fig 4. Classes of medicines used by interviewed cattle farmers.**

## Discussions

AMR is a public health issue that threatens the health of humans, animals, and the environment [7, 28]. Misuse of antimicrobials continues to be a major contributor to the development of resistance to disease-treating drugs [28]. Cattle farmers are the frontline defenders who make critical AMU decisions (prescription, dose, and administration). Farmers' AMU has been linked to their KAP, educational levels, views, and agricultural experiences [17, 29]. There is significant variance in cattle producers' access to, use of, and satisfaction with veterinary services among geographic areas, livestock production methods, animal health services, and social strata.

Livestock diseases are a major issue for cattle farmers in Rwanda and other low-income nations with large livestock populations and diverse climatic conditions that favor the presence of microorganisms. Antimicrobials are widely utilized to treat a variety of cattle diseases [2, 29]. The main reason for continued AMU reported by most farmers was treatment, with

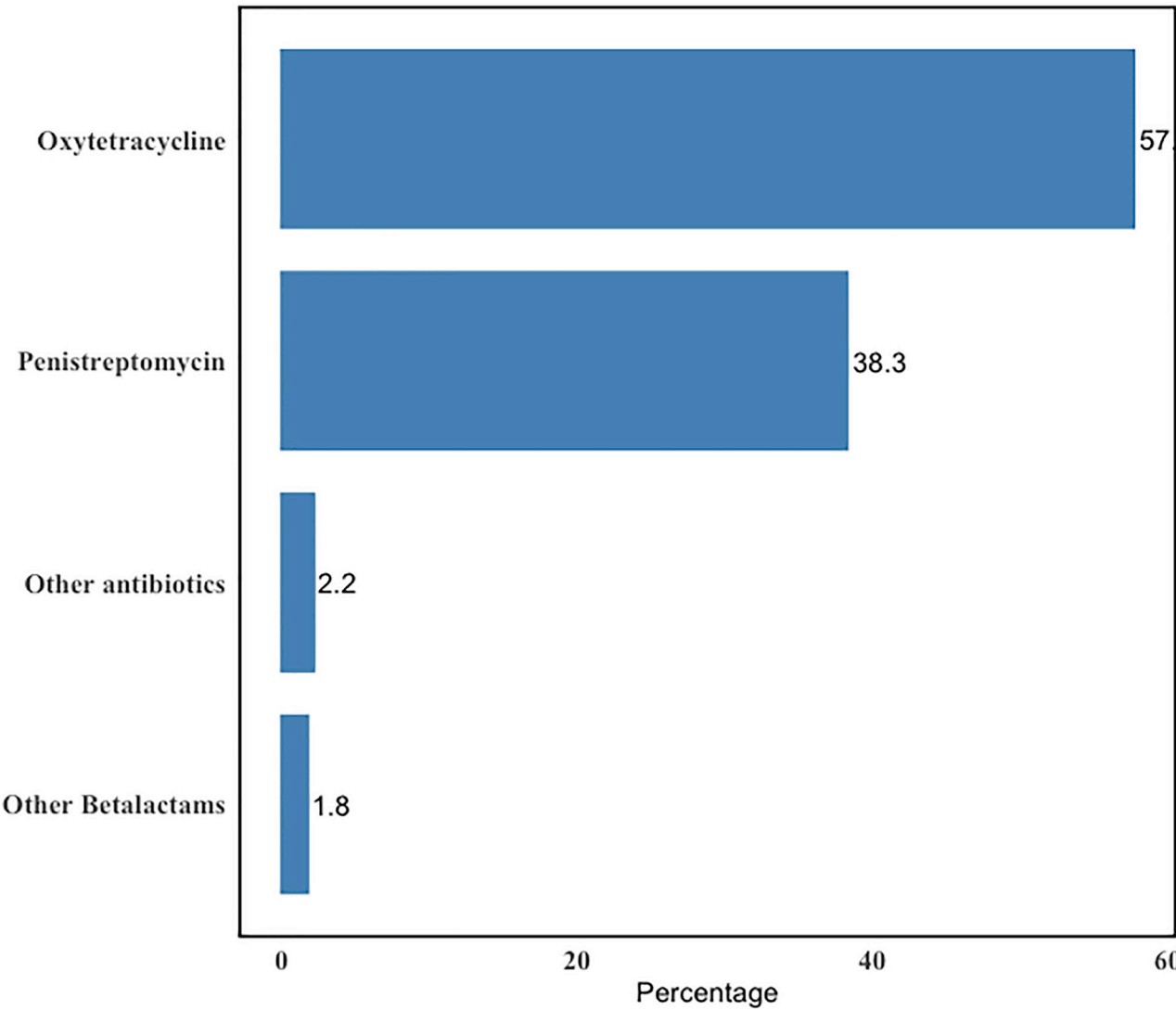

**Fig 5. Antibiotics commonly used by interviewed cattle farmers.**

infectious diseases caused by bacteria, parasites, fungi, and viruses being the main reported in the study areas. Gaps in biosecurity measures on farms and their linkages to AMU were observed in this study. Biosecurity measures include practices implemented on farms to prevent the introduction and circulation of pathogens. Consequently, implementing biosecurity measures is known to reduce the occurrence of infectious diseases on farms and could potentially minimize antimicrobials [30]. Water availability in farms was low with the majority having no water on farms (43.3%) and (60.1%) traveling more than 15 minutes to access clean water sources. The limited access to water may have contributed to most farms using sweeping as their cleaning method instead of washing with water and soap. Additionally, with the shortage of clean water in farms, cattle farmers are pushed to rear their cattle outside the farms in search of open-source drinking water. Such water sources are mostly contaminated by other farm animals' hooves and cow dung which has been a mode of transmission for infectious diseases such as anthrax and Foot Mouth Disease especially in dry areas in Africa [2, 31]. Our

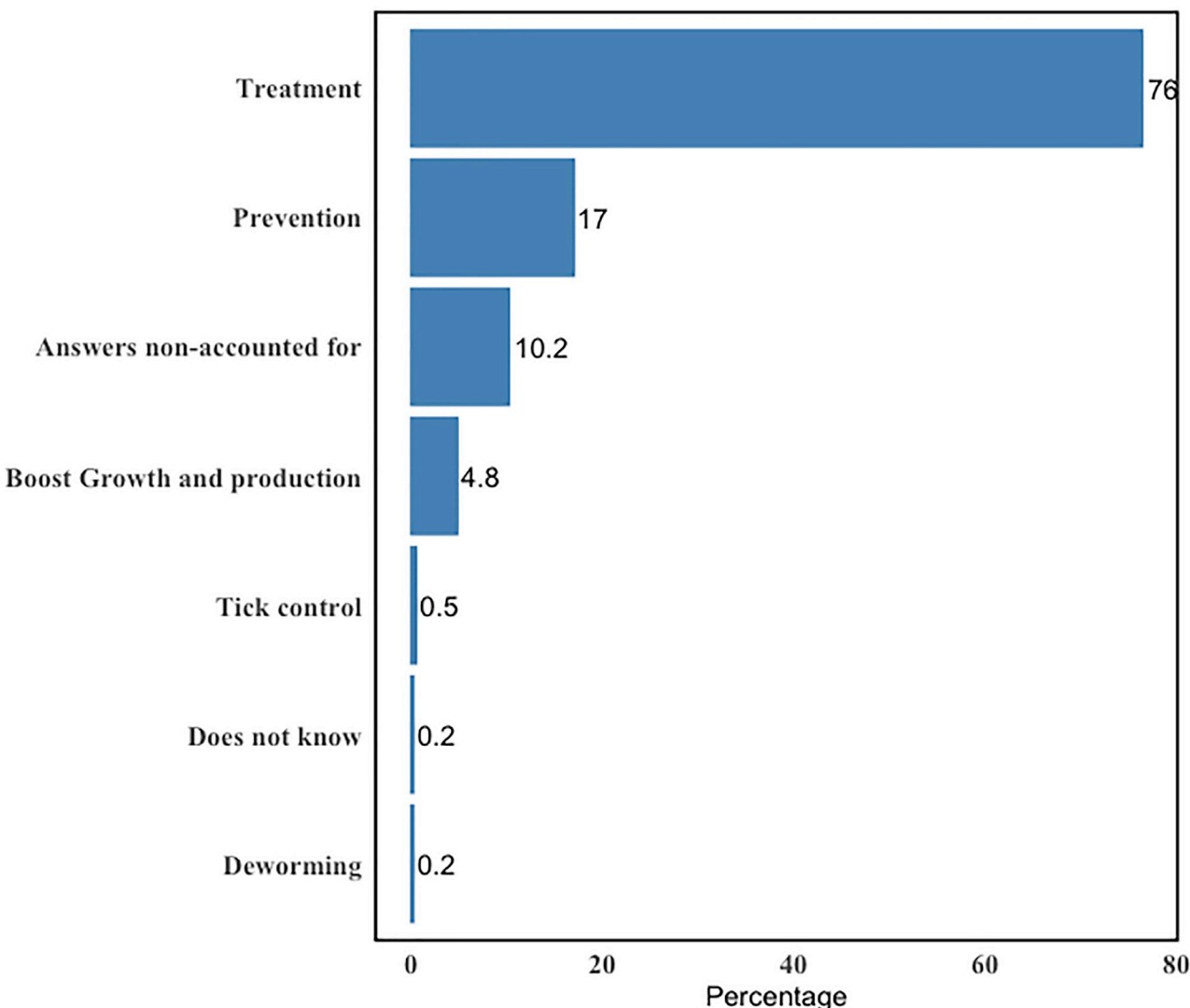

**Fig 6. Common reasons reported for AMU.**

study areas were classified as prone to dry seasons which can explain the water shortage in dry seasons, and more than half of our study participants practiced open grazing. These factors lead to inter-farm contamination and increase microbial disease prevalence perpetuating antimicrobial overuse [32]. Nevertheless, evidence shows that there is a correlation between farm biosecurity measures and levels of AMU; effective biosecurity measures are a leeway to reduce high levels of AMU on farms [32].

Good knowledge of vaccines, antibiotics, and antimicrobial resistance is expected to lessen the chances of antimicrobial misuse [33]. Most cattle farmers in this study do not comprehend that some diseases can acquire resistance to antibiotics, with more than half (52.8%) not knowing about AMR. Furthermore, 41.0% did not understand the effect of AMR whereas 21.8% of the participants reported that AMR makes it difficult to treat the cattle. More than a third of the participants did not understand the transmission mode of resistant pathogens which clearly shows that most farmers have no idea on the issue of AMR. The findings of this study

**Table 4. Knowledge-specific questions on AMR (N = 441).**

| Knowledge related items | Correct (%) | Incorrect (%) |
|---|---|---|
| Antibiotics are food supplement for livestock | 270 (61.2) | 171 (38.8) |
| Antibiotics are medicine for livestock | 387 (87.8) | 54 (12.2) |
| Antibiotics are both medicine and food supplement | 235 (53.3) | 206 (46.7) |
| Can antibiotics be useful for tick borne diseases? | 137 (31.1) | 304 (68.9) |
| Can antibiotics show side effects after use? | 88 (20.0) | 353 (80.0) |
| Can antibiotics be useful for weight gain in animals? | 141 (32.0) | 300 (68.0) |
| What do you think are impacts of antibiotics resistance? | 47 (10.7) | 394 (89.3) |
| Can bacteria develop resistance to antibiotics? | 239 (54.2) | 202 (45.8) |
| Humans can have the same pathogen causing diseases as animals | 195 (44.2) | 246 (55.8) |
| What do you think are modes of transmission for resistant pathogens from animals to humans? | 276 (62.6) | 165 (37.4) |
| **Categorical Level of Farmers' Knowledge (cutoff = 46%)*** | | |
| Good (≥ 46) | 238 (51.7) | |
| Poor (< 46) | 203 (48.3) | |

*The cutoff (46%) is the average knowledge score

are similar to reports from Ethiopia and other low-income countries in Africa [34]. Such low levels of knowledge have been the origin of exacerbated misuse of antimicrobials: i) using unprescribed animal medications, ii) using medicines for disease prevention and growth promotion [2]. This also cascades into malpractices such as eating dead carcasses caused by overdose, milking cattle that are on antimicrobials, and getting antimicrobials from friends/neighbors, among others. The same practices were reported in a study done in a similar setting in Maasai pastoralists [3]. Improved veterinary services and outreach would be instrumental in addressing these knowledge lacunae through contextualized/local antimicrobial stewardship programs and better biosecurity measures at farm levels to reduce AMU subsequently AMR [4].

According to this study, cattle farmers' knowledge varied considerably by demographic parameters such as participants' sex, educational background, marital status, district, and

**Table 5. Questions on attitudes toward AMR and AMU (N = 441).**

| Attitude related items | Agree (%) | Neutral (%) | Disagree (%) |
|---|---|---|---|
| antibiotics resistance is a public health concern | 255 (57.8) | 172 (39.0) | 14 (3.2) |
| If the cattle is sick and the veterinarian gives them medication and they don't get better, I should treat them myself | 247 (56.0) | 70 (15.9) | 124 (28.1) |
| Withdrawal period should be observed to avoid antibiotics residues in food products | 276 (62.6) | 118 (26.8) | 47 (10.7) |
| antibiotics should be prescribed by veterinarians only | 323 (73.2) | 44 (10.0) | 74 (16.8) |
| Misuse of antibiotics can cause antibiotics resistance | 338 (76.6) | 85 (19.3) | 18 (4.1) |
| Use of antibiotics for a long period can cause antibiotics resistance | 240 (54.4) | 168 (38.1) | 33 (7.5) |
| Sales and distribution of antibiotics would be beneficial to reduce antibiotics resistance | 22 (5.0) | 68 (15.4) | 351 (79.6) |
| Enhanced control and restrictions of antibiotics use would reduce emergence of antibiotics resistance | 236 (53.5) | 105 (23.8) | 100 (22.7) |
| Antibiotics should only be used to treat animal diseases | 356 (80.7) | 55 (12.5) | 30 (6.8) |
| **Categorical Level of Farmers' Attitude (cutoff = 69%)** | | | |
| Positive (≥ 69) | 247 (56.0) | | |
| Negative (< 69) | 194 (44.0) | | |

69% is the average attitude score

**Table 6. Questions on practices on AMR and AMU.**

| Practice related items | Correct (%) | Incorrect (%) |
|---|---|---|
| I call a veterinarian when my cattle are sick | 324 (73.5) | 117 (26.5) |
| I ask a friend for advice on which antibiotics to use when my cattle are sick | 219 (49.7) | 222 (50.3) |
| I get antibiotics from a veterinary pharmacy | 307 (69.6) | 134 (30.4) |
| I get antibiotics from a veterinary technician | 167 (37.9) | 274 (62.1) |
| I get my antibiotics from a friend and neighbors | 370 (83.9) | 71 (16.1) |
| I treat my cattle using antibiotics until they get better | 229 (51.9) | 212 (48.1) |
| I treat my cattle using antibiotics until their course of antibiotics is complete | 177 (40.1) | 264 (59.9) |
| When my cattle is better, I keep the remaining antibiotics for future use | 411 (93.2) | 30 (6.8) |
| When I have one sick cattle, I share the antibiotics to not sick animals | 411 (93.2) | 30 (6.8) |
| What do you do when the cow being treated with antibiotics dies? | 133 (30.2) | 308 (69.8) |
| When your cow is on antibiotics, do you milk it? | 176 (39.9) | 265 (60.1) |
| Do you give antibiotics to your cattle to promote their growth? | 273 (61.9) | 168 (38.1) |
| Do you give antibiotics to your cattle to prevent diseases? | 237 (53.7) | 204 (46.3) |
| Categorical Level of Farmers' Practices (Cutoff = 60%) | | |
| Good ($\geq 60$) | 267 (60.5) | |
| poor ($< 60$) | 174 (39.5) | |

60% is the average practice score

number of animals held. The poor knowledge scores of the participants in this study could be attributed to sex, where females had more odds of having better knowledge levels compared to male counterparts. This concords with the study that showed that sex differences are contextual and correlate with other sociodemographic factors especially education level and socioeconomic levels, with females having relatively more knowledge than their counterparts [35]. This also coincides with the study from Bangladesh that revealed that men leaned more towards buying unprescribed animal drugs compared to women; similarly, women-owned farms were also more likely to use fewer antimicrobials in farms compared to men-owned ones [36–38]. This discrepancy could be attributed to factors such as: i) women have more access to information; a cross-sectional study in Thailand showed that women had 1.8 times more odds of accessing information on AMR than men [39], ii) women have more knowledge on antibiotics because they tend to use prescribed medicines and are more aware of efficacy of medicine compared to men [40].

Our study showed that more than a half of participants (56%) had positive attitudes towards AMU and AMR. This is higher than levels found in similar studies conducted in Ethiopia [18, 19]. Encouragingly, most farmers (73.2%) have positive attitudes towards having veterinarians prescribe the medications for animals, as they believe that veterinarians have the required knowledge on cattle diseases, antimicrobials, and AMR. This concords findings from similar setting in Eastern Ethiopia that showed that pastoralists understood the need for veterinarians' prescriptions [18]. Nevertheless, more than half of the participants (56%) agreed that they could treat their cattle themselves if the diseases did not get treated, indicating that most cattle farmers do not understand why antimicrobials fail to treat sick cattle. This was the same in studies that showed that farmers tended to view themselves as ethno-veterinarians and the existence of a tendency to increase the dose of antimicrobials if their cattle were not getting better [19, 41, 42]. Access to adequate information or advice from animal care providers on how to use antimicrobials will minimize inappropriate AMU on cattle farms. A study supports

**Table 7. Pearson's correlation between KAP scores.**

| Variable | Knowledge | Attitude | Practice |
|---|---|---|---|
| **Knowledge** | | | |
| Pearson's correlation | 1 | 0.368** | 0.080 |
| Sig. (2-tailed) | | < 0.001 | 0.093 |
| **Attitude** | | | |
| Pearson's correlation | 0.368** | 1 | 0.338** |
| Sig. (2-tailed) | < 0.001 | | < 0.001 |
| **Practice** | | | |
| Pearson's correlation | 0.080 | 0.338** | 1 |
| Sig. (2-tailed) | 0.093 | < 0.001 | |

**Correlation is significant at the 0.01 level (2-tailed).

**Table 8. Associations of the farmers' knowledge, attitudes, and practices with demographic characteristics.**

| Independent variables | Knowledge % Correct | p value | Attitude % Correct | p value | Practice % Correct | p value |
|---|---|---|---|---|---|---|
| **Age category** | | | | | | |
| 18–30 | 46.4 | 0.209 | 51.8 | 0.743 | 53.6 | 0.348 |
| 31–45 | 60.5 | | 59.3 | | 65.1 | |
| 46–65 | 52.5 | | 57.1 | | 62.7 | |
| > 65 | 47.1 | | 55.9 | | 60.3 | |
| **Gender** | | | | | | |
| Male | 58.7 | < 0.001 | 59 | 0.105 | 55.7 | 0.009 |
| Female | 40.6 | | 51.2 | | 68.2 | |
| **Education level** | | | | | | |
| No education | 42.9 | 0.003 | 46.2 | 0.013 | 59.7 | 0.588 |
| Primary | 50.8 | | 57.1 | | 59.2 | |
| Secondary & above | 66.7 | | 66.7 | | 65.5 | |
| **Socioeconomic status** | | | | | | |
| Category 1 | 41.3 | 0.299 | 50 | 0.686 | 58.7 | 0.87 |
| Category 2 | 54.1 | | 56.8 | | 59.6 | |
| Category 3 | 51.9 | | 56.6 | | 61.8 | |
| **Marital status** | | | | | | |
| Single | 41.2 | 0.001 | 47.1 | 0.124 | 51.5 | 0.047 |
| Married | 57.1 | | 59 | | 60.3 | |
| Widowed & separated | 33.9 | | 50 | | 73.2 | |
| **District** | | | | | | |
| Nyagatare | 56.6 | 0.247 | 56.6 | 0.001 | 56.6 | < 0.001 |
| Gatsibo | 51.4 | | 51.4 | | 51.4 | |
| Kayonza | 46.9 | | 46.9 | | 46.9 | |
| **Role of participant** | | | | | | |
| Owner | 52.8 | 0.264 | 56.4 | 0.653 | 61.9 | 0.13 |
| Worker | 45 | | 53.3 | | 51.7 | |
| **Number of cattle** | | | | | | |
| < 3 cows | 47.1 | 0.033 | 62.7 | 0.002 | 69.3 | < 0.001 |
| ≥ 3 cows | 57.4 | | 47.7 | | 49.7 | |

**Table 9. Logistic regression analyses of the independent factors associated with KAP AMR.**

| Variable | Category | Knowledge OR (95% CI) | Attitudes OR (95% CI) | Practices OR (95% CI) |
|---|---|---|---|---|
| **Gender** | Male | 1.00 (reference) ** | 1.00 (reference) *** | 1.00 (reference) |
| | Female | 1.850 (1.182–2.895) | 1.956 (1.232–3.107) | 0.950 (0.588–1.536) |
| **Educational Background** | Secondary and above | 1.00 (reference) ** | 1.00 (reference) ** | 1.00 (reference) |
| | No formal education | 2.615 (1.396–4.900) | 2.699 (1.443–5.049) | 1.895 (1.000–3.592) |
| | Primary | 2.145 (1.230–3.742) | 1.859 (1.070–3.229) | 1.752 (1.004–3.056) |
| **Marital status** | Widowed & separated | 1.00 (reference) | 1.00 (reference) | 1.00 (reference) |
| | Single | 1.733 (0.733–4.098) | 1.454 (0.628–3.368) | 1.762 (0.706–4.396) |
| | Married | 0.627 (0.321–1.226) | 0.938 (0.486–1.811) | 1.487 (0.707–3.129) |
| **District** | Kayonza | 1.00 (reference) | 1.00 (reference) * | 1.00 (reference) *** |
| | Nyagatare | 1.097 (0.550–2.191) | 2.287 (1.145–4.568) | 4.769 (2.261–10.058) |
| | Gatsibo | 1.002 (0.617–1.628) | 1.746 (1.061–2.875) | 4.944 (2.821–8.666) |
| **Number of cattle** | $\geq$ 3 cows | 1.00 (reference) | 1.00 (reference) | 1.00 (reference) |
| | < 3 cows | 1.476 (0.812–2.683) | 0.656 (0.363–1.188) | 0.746 (0.400–1.391) |

Hosmer-Lemeshow goodness-of-fit test statistic for knowledge: 4.558, $p = 0.714$

Hosmer-Lemeshow goodness of fit test statistic for attitude: 7.095, $p = 0.526$

Hosmer-Lemeshow goodness-of-fit test statistic for practice: 14.065, $p = 0.080$

* = $p < 0.05$,

** = $p < 0.01$ and

*** $p < 0.001$

the importance of drug dispensers and clinicians counseling cattle farmers about antibiotic residues and improving their awareness of sensible AMU [28].

Our study shows that the farmers' attitude score was significantly influenced by their degree of education. The findings are consistent with previous research, which found that farmers' greater educational level substantially boosted their AMR attitude score [41, 43]. Attitudes scores were also associated to the location of farms which reflect the same findings in Tanzania, these discrepancies might be attributed to education levels differences in rural vis a vis urban area [43]. Targeted and tailored AMU stewardship efforts could reduce inappropriate attitudes that exacerbate AMR on farm level.

Farmers' self-prescription practices, while they have inadequate knowledge about drug selection and disease etiology, could result in the administration of sub-therapeutic doses or the misuse of antimicrobials. In the current study, 73.5% call a veterinarian when their cattle get sick and 49.7% of the participants ask their friends for advice on the type of antibiotics to use when their cattle are sick. This study also shows that more than three-quarters of the participants (83.9%) get antibiotics from friends or neighbors to treat their sick cattle. In the worst scenario, 93.2% of them keep the remaining antibiotics for future use. Extensive antibiotic misuse may enhance selection pressure for the formation of AMR and have detrimental health and economic consequences. This was confirmed by a study conducted in Ghana, Kenya, Tanzania, Zambia and Zimbabwe on households and pastoralists keeping layers, cattle, sheep, and goats [2]. Access to antimicrobials without a prescription and fragmented control of AMU in animal production are the key causes of AMR, according to our study findings Furthermore, as previously documented by the study [32], cattle farmers ceased delivering antimicrobials to their animals before the specified period of therapy for a variety of reasons in the current study, which is comparable with the current study findings. Previous research in Ethiopia on farmers' knowledge and attitudes toward AMU and AMR demonstrated that farmers'

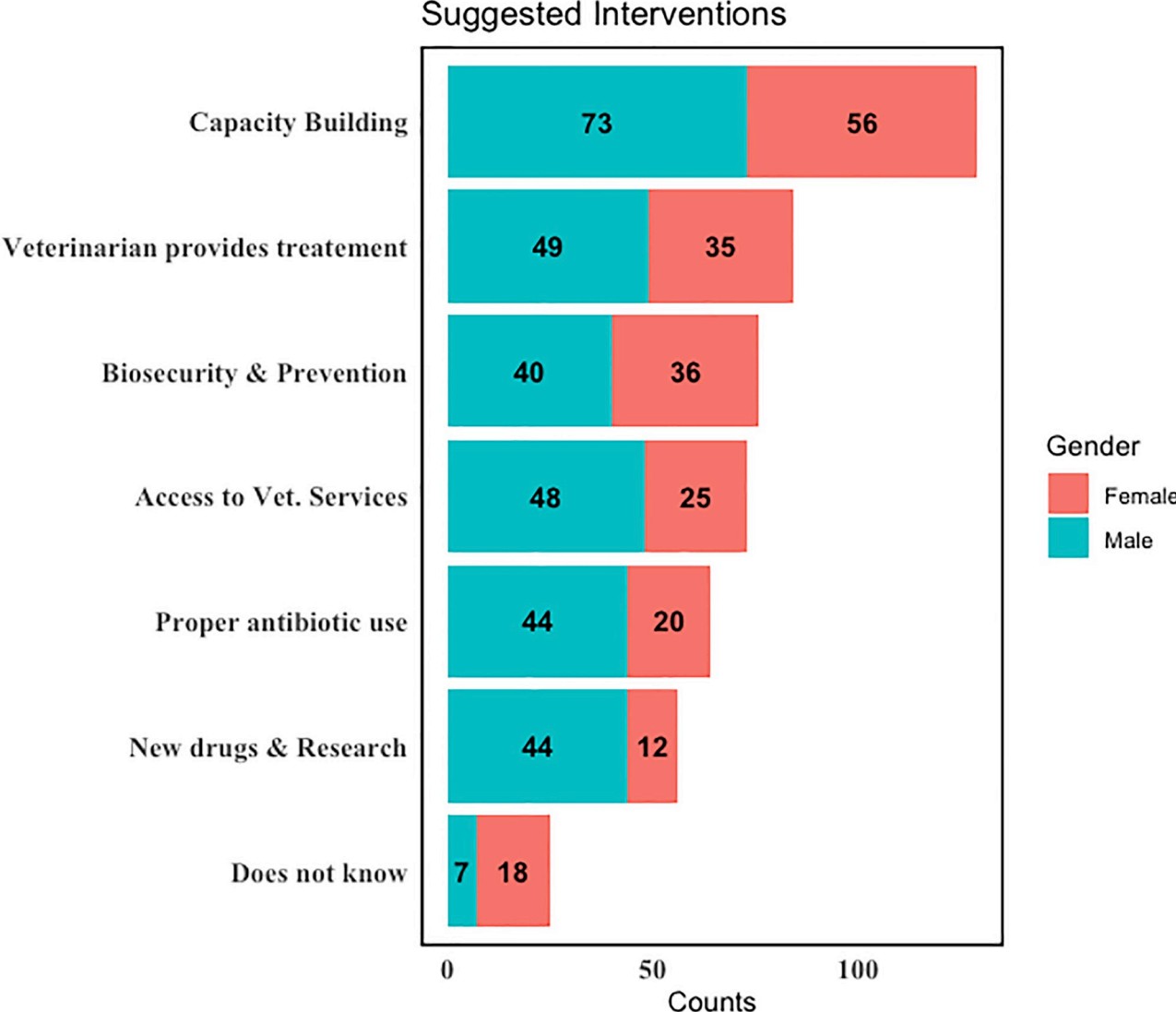

**Fig 7. Suggested interventions to reduce AMR.**

knowledge and attitudes are critical for influencing behavioral changes [18, 44]. Surprisingly, our research found a negative link between respondents with appropriate knowledge and desirable attitudes and poor AMU actions. This study discovered that farmers with a high degree of education and a substantially better awareness of antimicrobials had poor AMU behavior in cattle disease control. This might be because educated farmers and/or farmers with more years of farming experience have a better awareness of livestock illness and use available antimicrobials on sick animals themselves.

The use of antibiotics below the recommended dose and failure to adhere to treatment duration are two factors that contribute to AMR. Previous research has shown that people in the community have discontinued and altered the antibiotic dosage on their own initiative [19]. There was no question concerning low-dose antibiotic usage asked of the participants in this research. However, 59.9% of the participants reported that they did not utilize antibiotics until their course had elapsed. This result suggests that the farmers of cattle acted

independently in altering antibiotic dosages. In addition, 51.9% of the participants said they treated their cattle until they recovered, which showed that the recommended length of therapy was also not followed. There is little doubt that all of these incorrect applications might result in a rise in germs that are resistant to antibiotics.

Studies have found a strong relationship between AMU and other demographic factors including education level and place of residence [7–9]. In the current study, a statistically significant relationship between demographic factors including age and farm location and antibiotic awareness, use, and resistance was discovered. In Rwanda, those who work in animal husbandry tend to be older, less educated, and predominantly from rural regions. The study's findings supported this because 40% of the farmers who took part in it were at least 46 years old and 54% had only completed their primary education. In contrast to the current study's findings, investigations conducted in other countries found that the majority of farmers were younger and more educated [45]. As a result, it is considered that encouraging young individuals with a high degree of education to work in the livestock business will raise the perception and understanding of AMR.

## Conclusion

Our findings showed considerable differences in the KAP levels of participating farmers. For instance, gender, education level, and the district correlated with the level of knowledge, attitudes, or practices. They also highlighted improper behavior around AMU, such as farmers getting antimicrobials from friends and neighbors without any information on dosage or proper use from a veterinarian. These findings underscore the need for interventions to close gaps in the farmers' KAP of AMU and AMR while considering the different disparities among farmers in the study area. Therefore, AMR stewardship should entail a behavioral change to tackle the differences between knowledge, attitudes, and practices.

Additionally, given the participants' general low levels of KAP towards AMU and AMR, the competent authorities must strictly monitor veterinary pharmacies to restrict farmers' access to veterinary drugs. However, this overall supervision would only be possible through the increasing number of veterinary professionals to ensure proper regulation by experts at the community level.

In addition, other interventions should be implemented to enable access to clean water for use in farms and other biosecurity measures. Future research should explore AMR using laboratory experiments and include environmental aspects to fully understand AMR patterns within the Three Triads of One Health.

## Supporting information

**S1 Table. Questionnaire tool.**
(DOCX)

**S2 Table. Participants' demographics and farm characteristics.**
(XLSX)

**S3 Table. Biosecurity related answers.**
(XLSX)

**S4 Table. Antimicrobial use related answers.**
(XLSX)

**S5 Table. Knowledge related answers.**
(XLSX)

**S6 Table. Attitude related answers.**
(XLSX)

**S7 Table. Practice related answers.**
(XLSX)

## Acknowledgments

We are grateful to the enumerators Jean Bosco Nshutiyimana, Sylvestre Habimana, David Nibishaka, and Didier Habimana for their participation in data collection during this study.

## Author Contributions

**Conceptualization:** Anselme Shyaka.

**Data curation:** Elise M. Hirwa, Gisele Mujawamariya, Natnael Shimelash, Anselme Shyaka.

**Formal analysis:** Elise M. Hirwa, Gisele Mujawamariya, Natnael Shimelash, Anselme Shyaka.

**Investigation:** Elise M. Hirwa, Gisele Mujawamariya.

**Methodology:** Natnael Shimelash, Anselme Shyaka.

**Project administration:** Anselme Shyaka.

**Supervision:** Natnael Shimelash, Anselme Shyaka.

**Validation:** Gisele Mujawamariya, Natnael Shimelash, Anselme Shyaka.

**Visualization:** Anselme Shyaka.

**Writing – original draft:** Elise M. Hirwa, Gisele Mujawamariya, Natnael Shimelash, Anselme Shyaka.

**Writing – review & editing:** Elise M. Hirwa, Gisele Mujawamariya, Natnael Shimelash, Anselme Shyaka.

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
