## [Decision Letter · Decision Letter 0]

9 Jan 2024

PONE-D-23-36366Evaluation of cattle farmers’ knowledge, attitudes, and practices regarding antimicrobial use and antimicrobial resistance in RwandaPLOS ONE

Dear Dr. Shyaka,

Thank you for submitting your manuscript to PLOS ONE. After careful consideration, we feel that it has merit but does not fully meet PLOS ONE’s publication criteria as it currently stands. Therefore, we invite you to submit a revised version of the manuscript that addresses the points raised during the review process.

We look forward to receiving your revised manuscript.

Kind regards,

Raúl Alejandro Alegría-Morán, Ph.D.

Academic Editor

PLOS ONE

Journal Requirements:

Additional Editor Comments:

Please put particular emphasis on the level of detail of material and methods, more specifically regarding the statistical analysis.

Reviewers' comments:

Reviewer's Responses to Questions

**Comments to the Author**

1. Is the manuscript technically sound, and do the data support the conclusions?

Reviewer #1: Yes

Reviewer #2: Partly

2. Has the statistical analysis been performed appropriately and rigorously? 

Reviewer #1: Yes

Reviewer #2: Yes

3. Have the authors made all data underlying the findings in their manuscript fully available?

Reviewer #1: Yes

Reviewer #2: Yes

4. Is the manuscript presented in an intelligible fashion and written in standard English?

Reviewer #1: Yes

Reviewer #2: No

5. Review Comments to the Author

Reviewer #1: This paper is well written, and the topic is relevant.

The use of the mean KAP score as a cutoff to differentiate good and bad, can be difficult for future reference. It can be logical for this specific study but overall difficult to make comparisons with other similar studies.

Table 2: Second and third items in the first column are interchanged, correction is needed.

“How long do you travel to reach a water source from your farm or stable”, need to be the second and “Where do you get the water you use in the farm from?”, the third point.

Reviewer #2: The study appears to be well conducted, with an appropriate sample size, and executed correctly. However, it is recommended to order the methodology from general to particular, and in each part, provide as many details as possible.

The statistical analysis is simple and appears to have been carried out well. However, it is essential that the statistical analysis be explained in the methodology and not in the results. In addition, the assumptions of the analyses and the names of the procedures in SAS need to be detailed.

Finally, the manuscript presents various but small errors in the writing, so it is recommended to review. Some comments are found in the following lines.

ABSTRACT

Line 38. It is suggested not to use acronyms in keywords. Additionally, Antimicrobial resistance and Antimicrobial Use are already in the keyword list. I also suggest changing the country in the keywords and using a word that contextualizes your work, such as cattle farming.

Line 68. I suggest rewriting the phrase "as well as in food animals and their products" to "and food animals and their products."

METHODS

Lines 77 to 85. In the Study Setting, I suggest being concise in saying where the study will be carried out and the characteristics of choosing those places rather than providing data on the organization of the country.

Lines 87-88. I suggest writing: "Map of the study area with the interviewed farm locations (red points) in the Kayonza, Gatsibo, and Nyagatare districts."

Line 93: Study population and sampling strategy: In which statistic program was the calculation of the sample of interviews carried out? It should be clarified.

Line 96: I suggest changing "with reference to" to "concerning."

103-104: I suggest changing to "Finally, Excel's RAND function randomly selected three cells per sector (18 cells) and three villages per cell (54 villages)."

107-127: It is preferable to summarize the six items of the survey in tables.

109-110: I suggest editing "The research team designed and reviewed the data collection tool internally for content, understanding, and relevance."

124: The sentence "This entailed questions on how the interviewees felt or thought about topics related to AMR" generates confusion, so I suggest removing

129: I think that "Data collection procedure" should be included at the beginning of "Data collection tool and validation" since it provides the context of the data collection. Then, you can continue with the survey's structure.

140-157: I consider it important, after establishing in which statistical program the work was carried out, to mention the SAS procedures or packages that were used to obtain the results (data summary, associations, etc.). On the other hand, I consider that the description of analyses that were carried out should be described in the same order in which the results are presented. For example, descriptive statistics are mentioned near the end of the statistical analysis, but it is the first result. On the other hand, line 299 mentions a logistic regression analysis, which should be explained in detail in data analysis, including the procedure used in SAS and dependent and independent variables. The same goes for Pearson on line 268.

RESULTS

In the results section, it is essential to explain in the text what all the tables contain before describing them. For example, on line 160, it would be essential to write: “The socio-demographic data is summarized in Table 1” and then write the description of the Table.

Another comment is that “Figure (Fig.)” is summarized, but not the word Table, so I suggest keeping both without abbreviation for the entire text.

165: Correct the title of Table 1: Socio-demographics (data or variables) of cattle farmers

In Tables 1 and 2, it is important to be clear in the titles of each column. There must be a header in the columns where the response levels are described. In the column where the number of farmers per level and their percentage are described, it must also have a header.

DISCUSSION

343-344: change "use of" to just "use".

351-152: Before these lines, it is stated that AMU is due to different diseases. Then biosafety with the AMU is mentioned. It would be important to make the link between diseases, lack of biosecurity, and AMU here.

394-395: Which studies?

420-423: It is important that when reference is made to other studies, the context of those studies is provided (For example, place and type of rancher)

Conclusion

456-457: In the sentence “Our findings showed a considerable discrepancy in the KAP levels of farmers in terms of gender, education level, district, and other sociodemographic characteristics,” it would be necessary to mention those discrepancies briefly.

464-469: In my opinion, I think that ..this part of the conclusion escapes the proposed objectives

6. PLOS authors have the option to publish the peer review history of their article (what does this mean?). If published, this will include your full peer review and any attached files.

Reviewer #1: **Yes: **Kebede Amenu

Reviewer #2: **Yes: **Daniel Cartes

---

## [Author Response · Author response to Decision Letter 0]

24 Jan 2024

We have provided the requested corrections and information. Further, an IRB approval is attached along with the other submitted files.

We would like to mention that figure one was created by authors using QGIS Ver. 3.20. The layers are freely accessible from https://www.diva-gis.org/datadown and they can be shared under CC-BY license 4.0.

We have updated the figure caption to include that information.

---

## [Decision Letter · Decision Letter 1]

28 Feb 2024

PONE-D-23-36366R1Evaluation of cattle farmers’ knowledge, attitudes, and practices regarding antimicrobial use and antimicrobial resistance in RwandaPLOS ONE

Dear Dr. Shyaka,

Thank you for submitting your manuscript to PLOS ONE. After careful consideration, we feel that it has merit but does not fully meet PLOS ONE’s publication criteria as it currently stands. Therefore, we invite you to submit a revised version of the manuscript that addresses the points raised during the review process.

Please take in particular consideration what was highlighted by reviewer 2 regarding this point: "You should avoid formulating sentences that are too short on lines 102 and 161 to 168. Also, on line 161 you must delete (")"

We look forward to receiving your revised manuscript.

Kind regards,

Raúl Alejandro Alegría-Morán, Ph.D.

Academic Editor

PLOS ONE

Journal Requirements:

Reviewers' comments:

Reviewer's Responses to Questions

**Comments to the Author**

1. If the authors have adequately addressed your comments raised in a previous round of review and you feel that this manuscript is now acceptable for publication, you may indicate that here to bypass the “Comments to the Author” section, enter your conflict of interest statement in the “Confidential to Editor” section, and submit your "Accept" recommendation.

Reviewer #1: All comments have been addressed

Reviewer #2: All comments have been addressed

2. Is the manuscript technically sound, and do the data support the conclusions?

Reviewer #1: Yes

Reviewer #2: Yes

3. Has the statistical analysis been performed appropriately and rigorously? 

Reviewer #1: Yes

Reviewer #2: Yes

4. Have the authors made all data underlying the findings in their manuscript fully available?

Reviewer #1: No

Reviewer #2: Yes

5. Is the manuscript presented in an intelligible fashion and written in standard English?

Reviewer #1: Yes

Reviewer #2: Yes

6. Review Comments to the Author

Reviewer #1: The revision this manuscript: Evaluation of cattle farmers’ knowledge, attitudes, and practices regarding antimicrobial use and antimicrobial resistance in Rwanda, acceptable

Reviewer #2: I am satisfied with responses to the comments. You should avoid formulating sentences that are too short on lines 102 and 161 to 168. Also, on line 161 you must delete (")

7. PLOS authors have the option to publish the peer review history of their article (what does this mean?). If published, this will include your full peer review and any attached files.

Reviewer #1: **Yes: **Kebede Amenu

Reviewer #2: **Yes: **Daniel Cartes

---

## [Author Response · Author response to Decision Letter 1]

29 Feb 2024

Response: We are sincerely grateful for you’re the reviewer’s time to read and evaluate our paper. We believe that the final comments and suggestions will improve the clarity of our paper. We have modified the short sentences identified by the reviewer. 

Line 101: The sentences were as follows: 

To account for missing data and invalid entries, 15% of the 383 participants were added to the required sample size, rounding to 441 participants. Participants were selected using a multistage sampling method.

The two sentences were combined and to read as follows: 

To account for missing data and invalid entries, 15% of the 383 participants were added to the required sample size, rounding to 441 participants, selected using a multistage sampling method.

Line 161 to 168: The sentences were modified to remove very short sentences. The paragraph reads as follows: 

Most participants were male (61.5%; Table 1) and the majority (40.1%) were aged between 46 and 65 years and the mean age was 47.7 years (SD = 16.4). More than half (54%) of the participants had a primary school education, and the vast majority (86.4%) were farm owners who owned between 1–3 cattle (62.1%) (Table 1).

The “ symbol on Line 162 was also deleted.

---

## [Editor Report · Decision Letter 2]

5 Mar 2024

Evaluation of cattle farmers’ knowledge, attitudes, and practices regarding antimicrobial use and antimicrobial resistance in Rwanda

PONE-D-23-36366R2

Dear Dr. Shyaka,

We’re pleased to inform you that your manuscript has been judged scientifically suitable for publication and will be formally accepted for publication once it meets all outstanding technical requirements.

Kind regards,

Raúl Alejandro Alegría-Morán, Ph.D.

Academic Editor

PLOS ONE

Additional Editor Comments (optional): We greatly appreciate the inclusion of the comments and suggestions made by the reviewers. It is a very good manuscript and the incorporation of these improved the overall quality and understanding of the work.
---

## [Editor Report · Acceptance letter]

1 Apr 2024

PONE-D-23-36366R2 

PLOS ONE

Dear Dr. Shyaka, 

I'm pleased to inform you that your manuscript has been deemed suitable for publication in PLOS ONE. Congratulations! Your manuscript is now being handed over to our production team.

Kind regards, 

on behalf of

Dr. Raúl Alejandro Alegría-Morán 

Academic Editor

PLOS ONE